# Targeting Periostin Expression Makes Pancreatic Cancer Spheroids More Vulnerable to Natural Killer Cells

**DOI:** 10.3390/biomedicines11020270

**Published:** 2023-01-19

**Authors:** Didem Karakas, Merve Erkisa, Remzi Okan Akar, Gizem Akman, Ezgi Yudum Senol, Engin Ulukaya

**Affiliations:** 1Department of Molecular Biology and Genetics, Faculty of Engineering and Natural Sciences, Istinye University, Istanbul 34010, Turkey; 2Istinye University Molecular Cancer Research Center (ISUMKAM), Istinye University, Istanbul 34010, Turkey; 3Pancreatic Cancer Research Center, Acibadem University, Istanbul 34752, Turkey; 4Department of Molecular Medicine, Aziz Sancar DETAE, Istanbul University, Istanbul 34093, Turkey; 5Department of Clinical Biochemistry, Faculty of Medicine, Istinye University, Istanbul 34010, Turkey; 6Department of Biology, Faculty of Science, Istanbul University, Istanbul 34134, Turkey; 7Flow Cytometry Laboratory, Istinye University, Istanbul 34010, Turkey

**Keywords:** pancreatic cancer, pancreatic stellate cells, 3D co-culture, CRISPR/Cas9, periostin, collagen, natural killer cells

## Abstract

Pancreatic cancer (PaCa) characteristically has a dense tumor microenvironment, which results in poor patient prognosis. Pancreatic stellate cells (PSCs) are the most abundant cells in the PaCa microenvironment and the principal source of collagen. Periostin, a matricellular protein, is produced specifically by PSCs and promotes the aggressiveness of PaCa cells by facilitating extracellular collagen assembly. Here, we aimed to decrease extracellular collagen assembly by suppressing periostin, thereby increasing the cytotoxic activity of natural killer (NK) cells. Periostin expression was suppressed in PSCs (called PSC-P) using CRISPR-Cas9. PaCa cells (BxPC-3) were co-cultured with PSC and PSC-P cells in a 3D environment to form tumor spheroids mimicking the tumor microenvironment. The extracellular collagen production of spheroids was evaluated by Masson’s trichrome staining. The cytotoxic activity of NK-92 cells was analyzed by flow cytometry and confocal microscopy via CD107a staining. Cell death in BxPC-3 cells was evaluated by measuring Annexin-V and PI positivity using flow cytometry. As a result, periostin suppression decreased extracellular collagen and increased the infiltration of NK-92 cells into spheroids, and induced cell death in PaCa cells. In conclusion, we suggest that periostin might be a therapeutic target for PaCa and further analysis is warranted using in vivo models for proof-of-concept.

## 1. Introduction 

Pancreatic cancer (PaCa) is a highly fatal disease with an overall survival time of only 4–6 months following diagnosis [1,2,3]. Despite advancements in the detection and treatment of PaCa, the 5-year survival rate still stands at 3% for metastatic disease [4]. 

PaCa is characterized by a rich and unique tumor microenvironment (TME) composed of immune cells, blood vessels, fibroblasts, and many other types of cells [5], and is considered the main contributor to the aggressive nature of this disease [6,7]. A growing body of evidence indicates that TME plays critical roles in metastasis, angiogenesis, drug resistance, and evading apoptosis in various solid cancers, including PaCa [8,9,10,11]. 

Pancreatic stellate cells (PSCs), a group of exocrine functional myofibroblasts specific to the pancreas, are one of the most dominant cell types in the TME of PaCa [12,13]. In homeostatic conditions, these cells remain quiescent and are characterized by the presence of vitamin A-containing lipid droplets in their cytoplasms and a low capacity to produce extracellular matrix (ECM) proteins [12,14]. During pancreatic injury or acute inflammation of the pancreas, these cells are activated, lose their lipid droplets, express alpha-smooth muscle actin (α-SMA) and become contractile and secretory [15]. Activated PSCs become capable of the intensive proliferation, migration, and production of ECM proteins such as collagen, laminin, and fibronectin [16]. In chronic pancreatitis and pancreatic ductal adenocarcinoma (PDAC), PSCs remain in their active state and secrete numerous growth factors, cytokines and chemokines resulting in the aggressive characteristics of cancer cells. These cells are also the main source of collagen in PaCa tumors and form dense and characteristic desmoplastic stroma [15,16]. This dense stroma creates a physical barrier, prevents the infiltration of cytotoxic immune cells into tumors, and diminishes the delivery of chemotherapeutic drugs. 

Periostin is an extracellular matrix protein that contributes to bone development, tissue remodeling and repair, and tumorigenesis [17]. The overexpression of the periostin gene has been detected in various cancer types including pancreatic cancer [18,19,20,21,22]. In pancreatic cancer, periostin mRNA levels were shown to be 42 times higher than the normal pancreas [15] and PSCs are the only source of periostin in the pancreas [22,23]. Periostin has critical functions in ECM organization, particularly in collagen assembly [24], and serves as a scaffold for BMP-1 and collagen to accelerate collagen cross-linking [25]. It has two additional domains, allowing it to interact with other cells and some ECM proteins. Because of all these properties, periostin is considered one of the important proteins regulating ECM organization, especially collagen, in the PaCa tumor microenvironment and thereby influences the characteristics of cancer cells.

Natural killer (NK) cells are a group of innate immune cells that have spontaneous cytolytic activity against tumor cells and virus-infected cells [26]. Unlike T cells, NK cells do not require pre-sensitization or human leukocyte antigen (HLA/MHC) engagement to kill infected or malignant cells. Because of these unique properties, NK cell-mediated therapeutic approaches have emerged as promising therapeutic approaches, especially in hematological tumors and some types of solid tumors. However, the use of therapeutic approaches based on immune system cells in the treatment of PaCa [27,28] have shown limited success. The dense TME of PaCa is considered mainly responsible for the ineffectiveness of therapeutic approaches based on cytotoxic immune cells. The PaCa microenvironment provides a physical barrier to the penetration of effector T cells but not to immunosuppressive cells, such as myeloid-derived suppressor cells and M2-macrophages. 

Although studies have indicated that tumors with denser extracellular collagen are insensitive to the infiltration of cytotoxic T-cells [29], no study exists showing the effect of periostin on NK cells’ infiltration and their cytotoxic activity. Therefore, here we reduced collagen production in PaCa spheroids by suppressing periostin protein expression and then stimulated the cytotoxic activity of NK cells with an immunomodulatory drug (lenalidomide). Through this two-step approach, we aimed to increase the cytotoxic activity of NK cells against PaCa cells. 

For this purpose, BxPC-3 pancreatic cancer cells and PSCs were cultured as three-dimensional spheroids for modeling TME. Periostin was then suppressed in PSCs by CRISPR-Cas9 technology. The cytotoxic activity of NK cells was stimulated by treating them with lenalidomide. Spheroids were co-cultured with NK cells to assess the cytotoxic activity of NK cells and cell death in cancer cells. The results showed that decreased extracellular collagen levels after periostin suppression; reduced levels of collagen increased the cytotoxic activity of NK cells and cell death increased in both cancer cells and PSCs. In conclusion, targeting stroma and then stimulating the activity of NK cells may offer promising new treatment strategies for patients with PaCa. 

## 2. Materials and Methods 

### 2.1. Cell Lines and Cell Culture Conditions 

NK-92 natural killer cells were obtained from the American Culture Collection (ATCC) (Manassas, VA, USA). BxPC-3 human pancreatic cancer cell line were kindly gifted by Dr. Konstantinos Dimas (University of Thessaly, Greece). Human pancreatic stellate cells (PSCs) were obtained from ScienCell Research Laboratories (CA, USA). 

BxPC-3 and PSC cells were cultured in Dulbecco’s Modified Eagle’s medium (DMEM)/F12, supplemented with 10% fetal bovine serum (FBS) and 100 U/mL penicillin-streptomycin (P/S) solution. NK-92 cells were grown in Roswell Park Memorial Institute (RPMI)-1640, supplemented with 20% FBS, 10 ng/mL human recombinant IL-2 and 100 U/mL P/S solution. The cells were grown in 25 cm^2^ or 75 cm^2^ cell culture flasks at 37 °C in a humidified atmosphere containing 5% CO_2._ All cells were tested for the absence of mycoplasma. 

### 2.2. Three-Dimensional (3D) Cell Culture 

BxPC-3 and PSC cells were grown as 3D monoculture/co-culture spheroids on Poly(2-hydroxyethyl methacrylate)-HEMA coated plates. Poly-HEMA solution was prepared by dissolving Poly-HEMA (2% final) in an appropriate volume of 95% ethanol overnight on a heated stirrer at 60 °C. Then, plates were coated with Poly-HEMA solution (1000 μL for per wells of 6 well plates) and allowed to dry completely for 24 h at room temperature in a laminar flow cabinet. The plates were washed with PBS before seeding the cells. 

For 3D mono-culture spheroids, we seeded BxPC-3 or PSC cells at a density of 6 × 10^5^ cells/well in Poly-HEMA-coated 6 well plates. We used 1:1 ratio of BxPC-3 and PSCs to form co-culture spheroids (3 × 10^5^ cells/per cell type; 6 × 10^5^ in total). We incubated the plates at 37 °C for 4 days to allow forming spheroids. Then, the spheroids were incubated with NK-92 cells at a E:T ratio of 5:1 for 24 h. 

### 2.3. Labeling the Cells with CellTrackers 

Cell tracking permits the study of cell-cell interactions in heterogeneous mixtures, so the cells were labeled with non-toxic, stable cell tracking dyes, either CellTracker™ Blue CMAC or Green CMFDA Dye (Invitrogen, Waltham, MA, USA). Briefly, the cells were trypsinized and centrifuged at 300× *g* for 5 min. The supernatant was discarded, and the cell pellet was resuspended in complete culture media (DMEM:Ham’s F12+10% FBS) and CellTracker blue or green dye added at 10 µM final concentration and incubated at 37 °C for 30 min. The cells were then centrifuged at 300× *g* for 5 min, washed with PBS and centrifuged again at 300× *g* for another 5 min. The supernatant was discarded, and the pellet was resuspended with complete media, and mono-culture/co-culture spheroids were generated as described above. After 4 days of incubation, the cells were examined under fluorescent microscope to observe homotypic/heterotypic cell interaction.

### 2.4. Histological Staining for Extracellular Collagen in Spheroid Sections 

For collagen staining, paraffin-embedded spheroid sections first were prepared by following and modifying the protocol published by Shoval et al., 2017; Clayton et al., 2018 [30,31]. Briefly, spheroid suspensions in media were collected and centrifuged at 300× *g* for 5 min. The supernatant was discarded and the pellet was washed with PBS and centrifuged at 300× *g* for another 5 min. Then, the pellet was fixed with 4% paraformaldehyde (PFA) for 15 min at RT and washed with PBS to remove PFA. For the embedding process, the fixed spheroid pellets were suspended with 2% agarose. The agarose containing spheroids was pipetted to empty 6-well plates as small drops, and the drops were allowed to solidify for at least 1 h at room temperature. The PFA-fixed drops were then dehydrated in an ascending alcohol series (70% to 100% isopropanol in water), then left in xylene for 15 min. Samples were further processed two changes of xylene (15 min) followed by three baths of paraffin at 58 °C, and then embedded in a mold with paraffin. Sections were cut at 5 μm and mounted onto glass slides covered with Mayer’s albumin. 

The slides were deparaffinized and rehydrated through incubation in xylene and in a series of graded alcohols until water is used (ranging from 100% to 50% isopropanol). The slides were stained with trichrome dye using Masson’s Trichrome Staining kit according to manufacturer’s instructions (Bio-Optica, Milano, Italy). 

### 2.5. Suppression of POSTN Gene by CRISPR/Cas9

PSC cells were cultured at a density of 1 × 10^4^ cells in a 12-well culture plate for 24 h, then infected with LV-Cas9 lentivirus (Virostem Biotechnology, Turkey) containing either three different non-overlapping control small gRNAs (sgRNAs) or three periostin (POSTN) gRNAs at a MOI of 10 for 24 h. After 24 h, the media was replaced with fresh culture media and puromycin (2 µg/mL) was added for selection. Polybrene (5 µg/mL) was used to increase the efficiency of transfection. The media was replaced once every three days and selection process was complete on day 10. The establishment stable POSTN-suppressed cells were confirmed by Western blotting. 

### 2.6. Flow Cytometry Analysis 

For flow cytometry analysis, cells were cultured mono/co-culture spheroids, and then collected and trypsinized to obtain a single-cell suspension. Briefly, cells were washed in FACS buffer (PBS supplemented with 2% FBS) and, stained according to the manufacturer’s instructions in 100 µL FACS buffer supplemented with APC-conjugated anti-EPCAM (1:50, BD Biosciences, San Jose, CA, USA), PerCP-Cy 5.5-conjugated anti-CD56 (1:100, BDBiosciences, USA), PE-conjugated anti-CD107a (1:100, BD, Biosciences, USA) and APC conjugated anti α-SMA (1:2000, R&D Biotechne, Minneapolis, MN, USA). Cells were kept in the dark at RT for 20 min. Live/Dead cell analysis, Annexin-V-FITC (1:100, Cell Signaling Technology, Danvers, MA, USA) and Propidium Iodide-PE (1:10, Cell Signaling Technology, USA) were used. Stained cells were washed twice in DPBS to remove unbound antibodies, resuspended in 200 µL FACS buffer and analyzed in BD FACSCalibur Flow Cytometer (BD Biosciences). An unstained negative control was run to establish the fluorescence gates. First, the cells were gated in an FSC-A (forward scatter) and SSC-A (side scatter) dot plot to eliminate doublets. EPCAM (+) cells were chosen to analyze dead/live ratio in cancer cells. In order to identify CD107a (+) NK-92 cells, the CD56 (+) cell population was gated. All data were then analyzed using FlowJo X software Version 9. 

### 2.7. Immunofluorescent Staining of Spheroids 

For immunofluorescent analysis of spheroid cultures, cells were fixed with 4% PFA (Sigma-Aldrich, St. Louis, MO, USA) at RT for 15 min, washed three times with PBS, and blocked with 3% BSA in PBS. Following the blocking step, PE-conjugated anti-CD107a (1:50, BD Pharmingen, San Diego, CA, USA) primary antibody were applied to stain NK cells. Then the cells were washed three times and incubated with DAPI (250 nM, Biolegend, San Diego, CA, USA) at RT for 15 min to observe nuclei morphology. Cells were coverslipped with mounting medium (Dako) then observed under a confocal microscope (Leica). Immunofluorescence of PSC cells was analyzed as described above. Depending on the targeted proteins such as intracellular antigens, cells were permeabilized before blocking with 0.1% Triton X-100 (Sigma) to detect anti-Vimentin (1:1000, Cell Signaling Technology, USA) and anti-α-SMA (1:1000, R&D Biotechne, USA).

### 2.8. Western Blot

The cells were washed twice in ice-cold PBS and lysed in 1X RIPA buffer (Thermo Scientific) containing protease and phosphatase inhibitors for 30 min at 4 °C. Lysates were centrifuged at 13,000× *g* for 10 min at 4 °C and supernatants were collected. Total protein concentration was measured using the Pierce BCA protein assay kit (Thermo-Fisher Scientific, Waltham, MA, USA) and lysates were resuspended in Laemmli loading buffer (Bio-Rad, Hercules, CA, USA) and heated at 95 °C for 5 min. Equal amounts of protein (30–40 µg protein/lane) were subjected to SDS-PAGE with a 10% concentration for protein separation and electrotransferred to polyvinylidene difluoride (PVDF) membranes. The membranes were blocked with 5% dry milk in TBS containing 0.1% Tween-20 (TBS-T) and probed with periostin primary antibody, then labeled with horseradish peroxidase-conjugated anti-mouse secondary antibody (Cell Signaling Technology, MA, USA). The protein band was visualized using Western Blotting Luminol Reagent (Santa Cruz Biotechnology). The ImageQuant LAS 500 imaging system (GE Healthcare, Chicago, IL, USA) was used to detect the bands, and Fiji software was used to quantify the protein bands. GAPDH (Cell Signaling Technology, MA, USA) was used as the loading control. 

### 2.9. Statistical Analysis 

Data were expressed as mean or fold changes ± standard deviations (SDs). Student t-test determined statistical significance and GraphPad Prism 9.0 statistical software was used for IOS. *p* values indicate the probability of the means compared, being equal with * *p* < 0.05, ** *p* < 0.01, *** *p* < 0.001, **** *p* < 0.0001.

## 3. Results 

### 3.1. Periostin Is Specifically Expressed by Pancreatic Stellate Cells and Its High Expression Is Associated with Poor Patient Prognosis 

Pancreatic stellate cells (PSCs) are one of the most abundant cells in the PaCa tumor microenvironment and the principal source of collagen. Here, we first formed 3D cell spheroids by culturing the cells in poly-HEMA-coated plates to mimic tumor microenvironment-like conditions. Briefly, BxPC-3 pancreatic cancer cell line or PSCs was grown as a mono-culture spheroids and co-cultured at a 1:1 ratio for four days to produce 3D co-culture spheroids. Figure 1A shows the general morphology of BxPC-3 and PSC mono-culture spheroids and BxPC-3+PSC co-culture spheroids. To show heterotypic cell-cell interactions in co-culture spheroids we then labeled the cells with non-toxic CellTracker dyes. Briefly, BxPC-3 abd PSC cells were stained with blue and green dyes, respectively. The cells were then co-cultured for 4 days. As seen in Figure 1B, BxPC-3 and PSCs interacted with each other in the same spheroid structure, indicating the existence of heterotypic cell-cell interaction. 

Periostin is a matricellular protein responsible for extracellular collagen assembly and known to be specific for PSCs in the PaCa microenvironment. To demonstrate its specificity for PSCs, we measured periostin protein expression by Western blot. Figure 1C shows that BxPC-3 cells do not have periostin expression, while PSCs produce it in high amounts. Previous studies have indicated that high periostin expression might be an indicator of poor patient prognosis, so we analyzed the overall survival rates of PDAC patients with low and high periostin expression. Based on the data obtained by the KmPlotter database (Figure 1D), patients with high periostin expression have significantly shorter overall survival rates (*p* = 0.038).

### 3.2. Periostin Suppression Decreases a-SMA Expression and Collagen Production in PSCs 

In the next step, we aimed to knock out the periostin expression in PSCs using the CRISPR/Cas9 system. For this purpose, three different control, small guide RNAs (sgRNAs) and periostin sgRNAs (POSTN sgRNA) were designed and encapsulated in lentiviruses by Virostem Biotechnology. Among these three gRNAs, we chose the one causing the most periostin downregulation for further experiments. Briefly, PSCs were transduced with lentiviruses containing each sgRNAs at a multiplicity of infection (MOI) of 10 and selected with puromycin for 10 days. As shown in Figure 2A, high periostin expression was detected in the control sgRNA group, while a twofold decrease was found in periostin expression in the POSTN gRNA group. 

In pancreatic cancer, PSCs are able to change from a quiescent lipid-storing phenotype to a highly active α-SMA-expressing phenotype. We first evaluated whether the PSC cells used in the study were quiescent or active. Appendix A shows vimentin and α-SMA staining, indicating the presence of activated PSCs in culture conditions. Appendix A shows there were almost no lipid droplets in PSCs. In short, the cells used in the study were in their activated phenotype, which is associated with high periostin-producing ability. 

After periostin suppression, we evaluated the changes in α-SMA expression to determine the changes in the active status of the cells. We observed a strong periostin staining in the control sgRNA group and low positivity in the POSTN gRNA group (Figure 2B). This observation provides further evidence for the relationship between periostin inhibition and PSC activation. The PSC cells with low periostin and α-SMA expression were named PSC-P cells.

In the next step, we compared the changes in the migration and matrigel-degrading abilities of PSC and PSC-P cells. Both PSC and PSC-P cells were seeded in a matrigel drop at a density of 10,000 cells in 10 μL matrigel. At day 2 and 15, the invasive cells were evaluated and imaged. We observed that the invasion ability of PSC-P cells significantly decreased compared to PSCs (** *p* < 0.01) (Figure 2C).

Next, we co-cultured PSC or PSC-P cells with BxPC-3 cells. To show the presence of PSC cells in the spheroids, α-SMA positivity was analyzed by flow cytometry in both PSC mono-culture and BxPC-3+PSC co-culture spheroids (Appendix A). Then, we measured the size of spheroids formed by PSC or PSC-P cells as mono-culture or co-culture with BxPC-3 cells. As shown in Figure 2D, the size of spheroids in both mono-culture and co-culture conditions of PSC-P cells was detected to be significantly smaller than the spheroids formed by PSCs (* *p* < 0.05, ** *p* < 0.01). 

As periostin expression is associated with collagen assembly, we evaluated the changes in extracellular collagen. Briefly, spheroids were embedded in paraffin, then sectioned for the histological analysis of their collagen content and stained by Masson’s Trichome staining kit. As presented in Figure 2E, PSC mono-culture and BxPC-3+PSC co-culture spheroids have high extracellular collagen than the spheroids produced by either PSC-P or BxPC-3+PSC-P cells, as seen by their weaker blue/green staining.

### 3.3. Lenalidomide Treatment Increases the Number of CD107a-Positive Natural Killer Cells 

In the next step, we added NK-92 cells to spheroids to investigate whether their cytotoxic activity changes depending on the decreased collagen or not. After introducing NK-92 cells to spheroid cultures, we evaluated the physical interaction of the cells by labeling them with specific CellTracker dyes. BxPC-3 and PSC cells were stained with CellTracker red and green, respectively, and the cells were co-cultured in poly-HEMA coated plates for 4 days to form spheroids. NK-92 cells-labeled with CellTracker blue were added to spheroids to observe cell-cell interactions. Figure 3 shows the spheroids were covered by NK-92 cells following 24 h co-incubation.

Since the study aims to evaluate the effect of the changes in collagen by targeting periostin, and thereby assess the infiltration and activation of NK cells, we stimulated NK cells with the lenalidomide, which is a thalidomide analog. A dose range (0.15–20 μM) was selected for lenalidomide based on the literature, then NK-92 cells were treated with lenalidomide for 48 h to evaluate its effect on cell viability. The results showed that cell viability remains similar in all tested concentrations (Appendix A). 

The most commonly used two concentrations of lenalidomide (1 and 10 μM) were selected for further experiments. We pre-treated NK-92 cells with lenalidomide (1 and 10 μM) for 24 h and 48 h. Then, lenalidomide-stimulated NK-92 cells were added onto BxPC-3 mono-culture spheroids. The changes in CD107a expression (a marker for activated and cytotoxic NK-92 cells) were investigated by flow cytometry analysis. We detected an increased CD107a expression at 1 μM concentration for 24 h treatment (Figure 3B), so we selected a 24 h pre-treatment with 1 μM lenalidomide for further analyses. The quantitative graphs showing CD56 and CD107a status in NK-92 cells were given in Appendix A. 

### 3.4. Cytotoxic Activity of NK-92 Cells Increases in Spheroids Formed by PSC-P Cells 

To detect changes in the cytotoxic activity of NK-92 cells, we measured the percentage of CD107a-positive cells by flow cytometry. We first pre-treated NK-92 cells with 1 µM lenalidomide for 24 h, then transferred the cells onto the spheroids and incubated them for an additional 24 h. Following the incubation, we stained the cells with CD56 and CD107a antibodies, using CD56 antibody as a general marker to distinguish NK-92 cells from the total cell population. As shown in Figure 3C, the percentage of CD107a^+^ cells in CD56^+^ cell population was detected as 78.78% in BxPC-3 mono-culture spheroids. As expected, CD107a positivity were measured lower in BxPC-3+PSC co-culture spheroids (41.24%) than in BxPC-3 mono-culture spheroids. Consistent with our hypothesis and previous data, the percentage of CD107a^+^ cells increased in periostin suppressed group (BxPC-3+PSC-P) (62.35%) compared to BxPC-3+PSC spheroids (Figure 3C).

We used confocal microscopy to show the presence of CD107a^+^ NK-92 cells in spheroids. Briefly, spheroids groups were exposed to NK-92 cells as abovementioned. Then, the spheroids were stained with CD107a antibody and DAPI, and images were captured using a confocal laser scanning microscope (Figure 3D). Similar to flow cytometry analysis, the amount of cytotoxic NK-92 cells decreased in BxPC-3+PSC-P co-culture spheroids compared to BxPC-3+PSC spheroids (Figure 3D). These findings suggest that periostin plays an important role in creating collagen-rich environment that serves as a physical barrier to cytotoxic NK-92 cell infiltration.

### 3.5. NK-92 Cell-Mediated Cell Death Increases in Periostin-Suppressed Spheroids 

To analyze the cell death in total cell population, we evaluated Annexin-V and PI positivity by flow cytometry. Briefly, BxPC-3 mono-culture, BxPC-3+PSC and BxPC-3+PSC-P co-culture spheroids were stained by Annexin-V and PI following 24 h incubation with lenalidomide-stimulated NK-92 cells. Annexin-V and PI dyes were used to analyze early apoptotic, late apoptotic, and necrotic cells. Figure 4A shows the changes in the Annexin and PI positivity in different groups. The percentage of early apoptotic cells (Annexin-V positive and PI negative) decreased in the spheroids formed by periostin suppresses PSC cells (BxPC-3+PSC-P group) compared to BxPC-3+PSC group. We observed similar pattern in late apoptotic cells (Annexin-V and PI positive) (Figure 4A).

Having detected the enhanced cytotoxic activity of NK-92 cells against BxPC-3+PSC-P co-culture spheroids (Figure 3C,D), we also evaluated cell death in cancer cells. For this purpose, spheroids were exposed to NK-92 cells in the manner described. To detect cancer cells in the total population, cells were stained with EpCAM antibody. Following 24 h NK-92 cells exposure, 85.27% of BxPC-3 mono-culture spheroids was late apoptotic cells (Annexin-V and PI positive population) (Figure 4B). Consistent with CD107a analysis, the percentage of late apoptotic cells decreased in BxPC-3+PSC co-culture spheroids (70.23%). We detected an increase (88.32%) in the percentage of late apoptotic cells in the spheroids formed by periostin-suppressed PSC-P cells (BxPC-3+PSC-P) (Figure 4B). The quantitative graphs for Annexin-V and PI positivity in total cell population were given in Appendix A.

We also showed cell death in PSC and PSC-P cells in the spheroid structures. For this purpose, we labeled PSC and PSC-P cells with CellTracker Green. Then, we cultured the cells alone or with non-stained BxPC-3 cells for 4 days. Following 24 h exposure to NK-92 cells, we stained the cells with PI. As a result, PI positivity increased in PSC-P mono-culture spheroids compared to the spheroids formed by PSC cells (Figure 5A). Similar results were observed in co-culture spheroids formed by BxPC-3+PSC and BxPC-3+PSC-P cells (Figure 5B). 

## 4. Discussion 

Pancreatic cancer (PaCa) is one of the most aggressive and deadliest cancer types among all major cancers with 4–6 months of average survival time after diagnosis [1,32]. Despite new approaches for diagnosis and treatment, the overall survival (OS) and relapse-free survival (RFS) period of patients remains 15.4 months and 9.6 months, respectively [33].

PaCa is characterized by a lack of specific symptoms, exhibits early invasion and metastasis capabilities and resistance to chemotherapy and radiotherapy [34,35]. One of the main reasons underlying this aggressive nature of PaCa is its characteristic and dense tumor microenvironment (TME). In addition to cancer cells, TME consists of active fibroblasts, immune cells, adipocytes, blood and lymph vessels, and extracellular matrix. There is an active and dynamic relationship between cancer cells and the cells in TME, and this interaction contributes to all hallmarks of cancer [36].

Our study shows for the first time that targeting periostin and thereby decreasing collagen density increases the infiltration and the cytotoxic activity of natural killer cells against pancreatic cancer cells and pancreatic stellate cells.

Numerous studies have indicated that TME benefits cancer cells by supporting migration and invasion, contributing to drug resistance, and facilitating escape from apoptotic cell death [9,37,38,39]. In addition, the bidirectional interaction between cancer cells and the other cells in TME creates a tumor-supportive environment and prevents the infiltration of anti-tumorigenic immune cells (such as CD8+ cytotoxic T-lymphocytes or natural killer cells) into the tumor. Meanwhile, other cell groups of TME with pro-tumorigenic activity are allowed by tumor cells into the tumor to support their aggressiveness [40,41]. The TME is considered the major determinant of these critical events. 

Pancreatic stellate cells (PSCs), a group of fibroblast-like cells with exocrine functions, are one of the most abundant cell types in PaCa stroma. These cells usually exist in a quiescent state in the physiological conditions of a healthy pancreas. Quiescent PSCs characteristically have lipid droplets containing vitamin A, exhibit normal endocrine and exocrine secretion and maintain extracellular matrix composition [12,42]. PSCs can become activated during various processes including inflammation, injury, and cancer progression, resulting in a loss of their vitamin A-storing lipid droplets [12]. Besides, activated PSCs have an increase in their secretory phenotype, express alpha-smooth muscle actin (α-SMA) and large amounts of extracellular matrix (ECM) proteins and metalloproteinases to remodel the ECM structure [12,43,44]. In PDAC progression, active PSCs contribute to the aggressiveness of cancer cells by producing high amount of ECM proteins to create a dense stroma that acts as a physical barrier to infiltration of cytotoxic immune cells and delivery of chemotherapeutics to the tumor [45,46,47,48]. Further, α-SMA expression in tumors has been associated with PDAC clinicopathological characteristics and is considered an independent prognostic biomarker for PDAC patients [43,44,49]. 

The dense ECM of PaCa tumors is dominantly composed of collagen, hyaluronic acid, laminin, and fibronectin. Fibrillar collagens are the most abundant ECM proteins in PaCa stroma, accounting for more than 80% of total ECM mass [50]. Active fibroblasts and PSC cells produce the majority of collagen in the ECM [51], while cancer cells contribute only a limited amount [50]. In TME, collagen fibrils produced by PSCs/fibroblasts are stabilized through cross-linkages with lysyl oxidases (LOX) [52]. 

Periostin is an extracellular matrix protein that functions in bone development, tissue remodeling and repair, and tumorigenesis [17]. It plays an important role in ECM structure and organization by activating LOX enzymes to induce collagen cross-linking and assembly [24,25]. In addition, periostin can interact with the cells or ECM proteins through its specific domains on its N- and C-terminals. In vitro and in vivo studies reported that periostin expression is associated with the aggressive characteristics of cancer cells including proliferation, invasion and metastasis, angiogenesis, drug resistance and avoiding immune destruction [24]. Therefore, periostin expression is defined as a negative prognostic marker for the patients with various cancer types, including pancreatic cancer [18,53,54]. 

The PaCa microenvironment is defined as an immunosuppressive microenvironment characterized by an increased infiltration of regulatory T cells (Treg), myeloid-derived immunosuppressive cells (MDSC), and tumor-associated macrophages (TAM) [55]. On the other hand, the infiltration of cytotoxic immune cells, such as CD8+ T cells and natural killer cells (NKs), is known to be limited. Studies indicate that the dense stroma of PaCa is a main factor for weak infiltration of cytotoxic immune cells. Dense collagen in stroma not only deters the migration and infiltration of CD8+ cytotoxic T cells in the tumor, but also decreases their proliferation and cytotoxic activity [56]. A study performed with ex vivo models of lung cancer showed that collagen degradation caused the significantly increased tumor infiltration of T cells, enabling T cell-tumor cell communication resulting in cancer cell death [29]. Another study in breast cancer showed that the abundance of infiltrating T cells decreased in tumors with high collagen density [57].

Based on the information and findings, the aim of this study is to decrease extracellular collagen assembly by targeting periostin and induce the infiltration and cytotoxic activity of NK cells.

For this purpose, BxPC-3 pancreatic cancer cells were grown alone (mono-culture) and co-cultured with PSC cells on a low-attachment surface to induce the formation of three-dimensional (3D) tumor-mimicking spheroid structures. Cells grown in a 3D culture environment are known to have tissue-like physical and biochemical properties [58]. After the cells were grown as mono-culture and co-culture spheroids, they were labeled with specific fluorescent dyes (CellTracker) and tracked to see whether BxPC-3 cells and PSCs cells were located in the same spheroid structures or not. Briefly, we observed that BxPC-3 and PSC cells interact with each other and form 3D spheroids, indicating the presence of heterotypic cell-cell interactions. 

Then, we stained collagen protein in the spheroid structures using Trichrome Masson’s staining kit. Although this method is generally used to stain the tissue sections, studies showed that it can be adapted to stain the sections taken from spheroids [30]. Based on the collagen staining results, intense collagen staining was observed in PSC mono-culture spheroids. Similarly, we obtained positive staining for BxPC-3 mono-culture spheroids but this was weaker than PSC spheroids (data not shown). We also observed a staining pattern in BxPC-3+PSC co-culture spheroids similar to that of the PSC mono-culture structures. This is consistent with literature indicating PSC cells are the main source of collagen. 

Since we aim to suppress periostin protein, we first confirmed the specificity of periostin for PSC cells by performing a western blot. Our data was consistent with the data published earlier in the literature [22,23]. Then, we permanently suppressed periostin protein production by CRISPR/Cas9 technology. Western blot analysis revealed more than a 50% reduction in periostin protein expression compared to the control group. However, we could not delete the whole periostin protein in PSCs. A potential reason for this was discussed in Nature Methods in 2019. In the study, 136 different genes were targeted to knock out, but 1/3 of these genes had protein expression, even though it was lower than the original level. The authors explained the phenomenon with two causal mechanisms; translation reinitiation leading to N-terminally truncated target proteins, or skipping the edited exon leading to protein isoforms with internal sequence deletions [59]. 

Next, we stained the PSC and periostin-suppressed PSC cells (called PSC-P) with α-SMA antibodies to compare their active/passive states. A significant reduction in α-SMA positivity was detected in the periostin-suppressed group (PSC-P) compared to the control.

Since α -SMA, the marker of activated PSCs, was found to decrease in PSC-P cells, we also evaluated the differences in the invasion ability of spheroids formed by PSC or PSC-P cells by matrigel drop invasion analysis. Consistent with literature [60], we showed the number of invasive cells significantly reduced in PSC-P group compared to PSC cells. Similarly, the size of spheroids formed by either PSC-P mono-culture or co-culture with BxPC-3 cells were smaller than the spheroids formed by PSC cells. Excitingly, and consistent with literature, the collagen density was observed to dramatically reduce in the spheroids formed by PSC-P, providing further evidence of the validity of our hypothesis.

Our next step was to stimulate the cytotoxic activity of NK-92 cells by treating them with lenalidomide, a thalidomide analogue. Studies show that lenalidomide administration increases the cytotoxic activity of NK cells [61,62]. To determine optimal treatment conditions, NK-92 cells were treated with two different concentrations of lenalidomide for 24 and 48 h. Based on the results, 1 μM lenalidomide was selected and applied to cells for 24 h to stimulate NK-92 cells.

Because the aim of our study is to increase NK-92 cell-dependent cytotoxicity by targeting periostin to decrease collagen, we evaluated the infiltration of NK-92 cells into spheroids and their cytotoxic activity. We first pre-treated NK-92 cells with 1 μM lenalidomide for 24 h. Then we applied the pre-treated cells to spheroids formed by BxPC-3 mono-culture, and its co-culture with PSC or PSC-P cells. After 24 h incubation, we evaluated the activation of NK-92 cells by flow cytometry analysis of CD107a+ cells. There was almost no CD107a positivity in NK-92 cells in the absence of cancer cells or PSCs (data not shown). When NK-92 cells were cultured with BxPC-3 mono-culture, the percentage of CD107a-positive cells dramatically increased (78.78%). As expected, the number of CD107a-positive cells decreased in the co-culture spheroids formed by BxPC-3 and PSC cells (41.24%). Consistent with our collagen analysis, the cytotoxic activity of NK-92 cells was restored in co-culture spheroids generated by PSC-P cells (62.35%). 

To confirm flow cytometry analysis, we stained the spheroids with CD107a antibody and assessed the infiltration of cytotoxic NK-92 cells into spheroids using confocal microscopy. Similar to flow cytometry data, we detected that the infiltration of cytotoxic NK-92 cells into BxPC-3+PSC-P co-culture spheroids increased compared to BxPC-3+PSC spheroids. As we mentioned before, dense stroma provides a physical barrier for immune cells, preventing their cytotoxic action. Although collagen inhibition has been shown to increase the infiltration of cytotoxic T cells into tumors of various cancers [29,56,57], there is no study showing the effect of periostin on the infiltration of NK cells in the literature. Here, we showed for the first time that the inhibition of periostin increased the infiltration of cytotoxic NK-92 cells into tumor-mimicking spheroid structures. 

Furthermore, we assessed cell death in PaCa cells by flow cytometry analysis of Annexin-V and PI positivity. We observed the same pattern as with CD107a staining. Briefly, the number of late apoptotic cells in BxPC-3+PSC-P co-culture spheroids were detected to be higher than BxPC-3+PSC spheroids (88.32% vs. 70.23%, respectively).

In addition to cancer cell death, we aimed to detect cell death in PSC cells by PI staining. We labeled PSC cells with CellTracker Green, leaving BxPC-3 cells unstained. Then, we stained the cells with PI after 24 h incubation with lenalidomide-treated NK-92 cells. Briefly, similar to flow cytometry analysis, PI positivity in BxPC-3+PSC-P spheroids was higher than in the BxPC-3+PSC group. 

Although there are no clinical trials targeting periostin currently, some ongoing or completed clinical trials exist focusing on pancreatic cancer stroma. A meta-analysis performed by searching MEDLINE/PubMed and the EMBASE database reported that three studies were found to focus on targeting hyaluronic acid in pancreatic cancer stroma [63]. The treatment of pancreatic cancer patients with hyaluronidase (PEGPH20) did not cause promising patient survival. Therefore, PEGPH20 was used in combination with gemcitabine, nab-paclitaxel, or FOLFIRINOX, which are the main treatment options for pancreatic cancer patients. In a phase Ib/II clinical trial study, the combination of PEGPH20 with gemcitabine and nab-paclitaxel was tested on stage IV pancreatic cancer patients [64]. The progression-free survival (PFS) of the patients was detected to increase from 6.3 months to 9.2 months [64]. A second phase II trial from the same group reported that the overall survival (OS) of the patients was found to be 6 months in pancreatic cancer patients with high hyaluronic acid expression [65]. On the other hand, the results obtained from the combination of PEGPH20 with FOLFIRINOX were not promising and the trial was closed in 2017. The median OS for treatment with FOLFIRINOX was reported at 14.4 months and with PEGPH at only 7.7 months [66]. 

The effect of another potential stroma-targeting drug candidate, a Hedgehog pathway inhibitor (IPI-926), was tested in both pre-clinical models and clinical trials. In pre-clinical studies, IPI-926 decreased desmoplasia and increased the delivery of gemcitabine to tumors [67]. However, in the clinical trials, the combination of IPI-926 with gemcitabine caused an increase in progressive disease compared to gemcitabine alone [68]. As seen in this example, our understanding tumor-stroma interaction is still too limited and we need further studies to elucidate these complex events that promote cancer aggressiveness.

To conclude, this study is the first showing the impact of periostin on the infiltration of NK cells into 3D spheroid structures generated by PaCa cells and PSC cells. Our data identified periostin as a potential target to increase the infiltration of cytotoxic NK-92 cells into spheroids. Cell death was detected to increase consistently in the periostin-suppressed group. We suggest that periostin may serve as a potential target to increase cancer cell-NK interaction resulting in the stimulation of cell death. On the other hand, some studies have reported that targeting stroma is a double-edged sword because its inhibition may induce cancer metastasis. Therefore, this should be taken into consideration while using stoma targeting approaches, and the results are needed to further support in vivo data. 

## Figures and Tables

**Figure 1 biomedicines-11-00270-f001:**
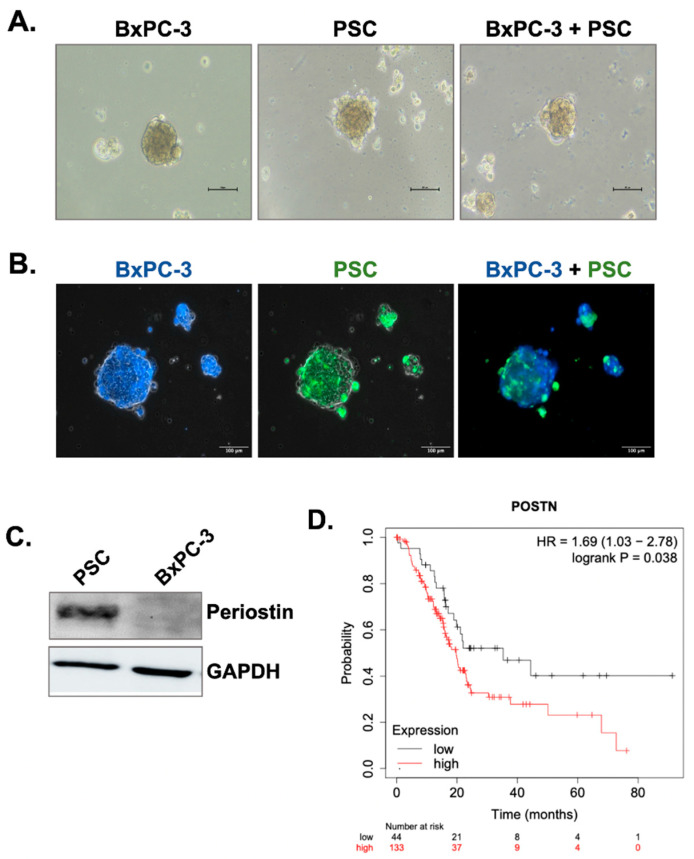
Periostin is specifically produced by PSCs and its overexpression is related poor prognosis in PDAC patients. (**A**) BxPC-3 cells were co-cultured with PSCs in poly-HEMA coated plates to form spheroids. The images were taken 4 days after seeding. (**B**) To show heterotypic cell-cell interactions in the same spheroid, BxPC-3 cells and PSC cells were labeled with CellTracker blue and green, respectively. The images represent the same spheroid area captured by specific filters for blue and green dyes and then merged. (**C**) Western blot analysis of periostin expression in BxPC-3 pancreatic cancer cells and PSCs. GAPDH was used as the loading control. (**D**) Kaplan-Meier survival analysis of PDAC patients with low and high periostin mRNA expression (*p* = 0.038).

**Figure 2 biomedicines-11-00270-f002:**
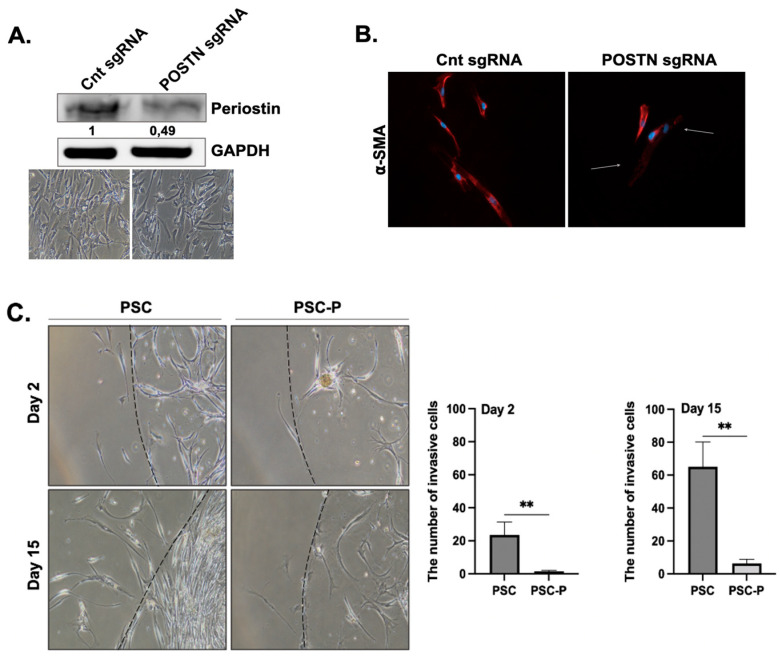
Periostin suppression in PSCs decreased their invasion, and reduced the size and extracellular collagen in spheroids. (**A**) The changes in periostin protein expression were evaluated by Western blot. The phase contrast images were added to show general cell morphology in both the control sgRNA and POSTN sgRNA group indicating there were no changes in cell morphology and viability after POSTN suppression. (**B**) α-SMA positivity of control and POSTN sgRNA groups were evaluated by immunofluorescent staining. Red; a-SMA staining, blue; DAPI staining. Arrows show decreased α-SMA positivity in POSTN sgRNA group. (**C**) The changes in the invasion ability of non-treated PSC cells and periostin-suppressed cells (named PSC-P) were assessed by matrigel drop analysis. Cells were seeded into matrigel drops and imaged on days 2 and 15. Graphs show the number of invasive cells. (**D**) The spheroid size was evaluated by measuring the spheroid diameter using Image J software (Fiji software, version 1.0). Upper images show the mono-culture spheroids of PSC and PSC-P cells. Lower images show co-culture spheroids generated by BxPC-3+PSC cells and BxPC-3+PSC-P cells. (**E**) The changes in collagen levels of spheroids were evaluated by Trichrome Masson’s staining. Blue/green staining indicates collagen positivity in spheroids. * *p* < 0.05, ** *p* < 0.01.

**Figure 3 biomedicines-11-00270-f003:**
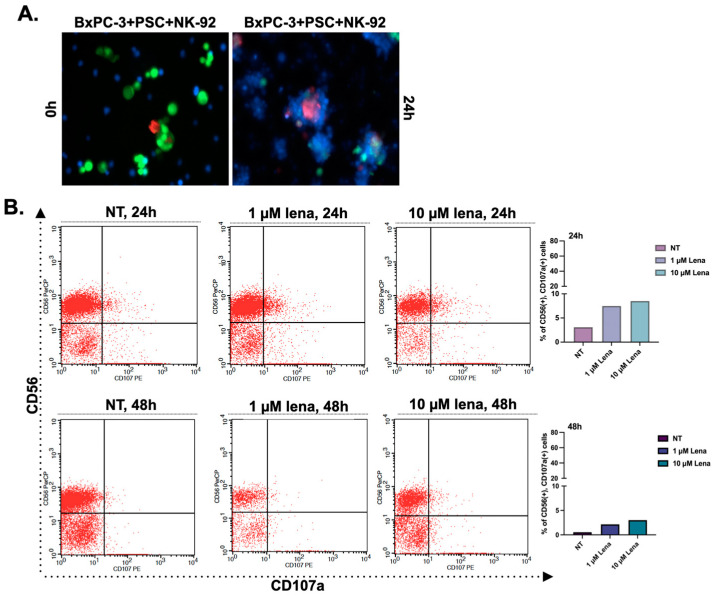
The cytotoxic activity of NK-92 cells increased in the spheroids formed by periostin-suppressed cells. (**A**) Cell-cell interactions in spheroids structures were observed by labeling the cells with CellTracker dyes. BxPC-3 and PSC cells were stained with CellTracker red and green, respectively. Then, the cells were co-cultured in poly-HEMA coated plates for four days to form spheroids. NK-92 cells with CellTracker blue were added to spheroids to observe cell-cell interactions. (**B**) NK-92 cells were treated with two different concentrations of lenalidomide (1 and 10 µM) for 24 and 48 h. NT group left untreated. Then the cells were cultured with BxPC-3 cells for 24 h to evaluate their cytotoxic activity. CD56 antibody was used to distinguish NK-92 cells from BxPC-3 cells. Then, CD107a^+^ cells were analyzed in the CD56^+^ population to assess the cytotoxic activity of NK-92 cells. Lena; lenalidomide. (**C**) BxPC-3 mono-culture, BxPC-3+PSC, and BxPC-3+PSC-P co-culture spheroids were exposed to NK-92 cells pre-treated with 1 µM lenalidomide for 24 h. Following 24 h exposure, the percentage of CD107a^+^ cells was analyzed in the CD56^+^ population. (**D**) Confocal images of spheroids cultured with lenalidomide-treated NK-92 cells for 24 h. Green shows the infiltration of CD107a^+^ NK-92 cells, and blue shows DAPI staining for nuclei. * *p* < 0.05, ** *p* < 0.01.

**Figure 4 biomedicines-11-00270-f004:**
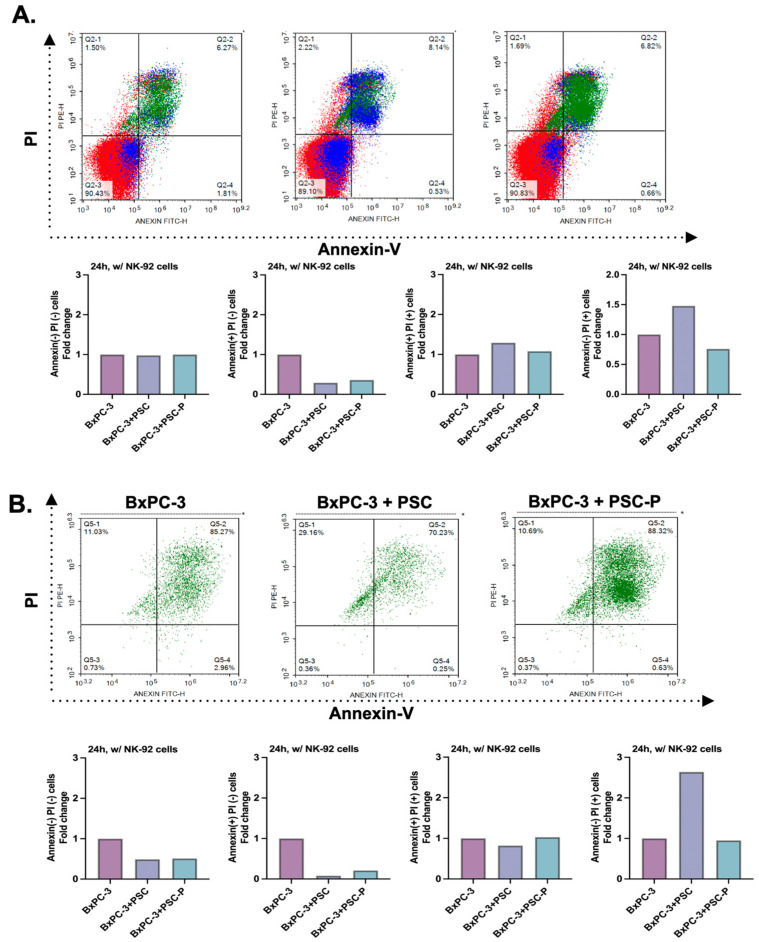
Cell death was detected to increase in periostin-suppressed spheroids. (**A**) Annexin-V and PI positivity in total cell populations. The graphs show the percentage of Annexin-V and PI positivity in BxPC-3 cells mono-culture, BxPC-3+PSC co-culture and BxPC-3+PSC-P co-culture spheroids following 24 h exposure to lenalidomide-stimulated NK-92 cells. (**B**) Annexin-V and PI positivity in EpCAM(+) cancer cell population. The graphs show the percentage of Annexin-V and PI staining in EpCAM(+) BxPC-3 mono-culture, BxPC-3+PSC, and BxPC-3+PSC-P co-culture spheroids were exposed to lenalidomide-stimulated NK-92 cells for 24 h.

**Figure 5 biomedicines-11-00270-f005:**
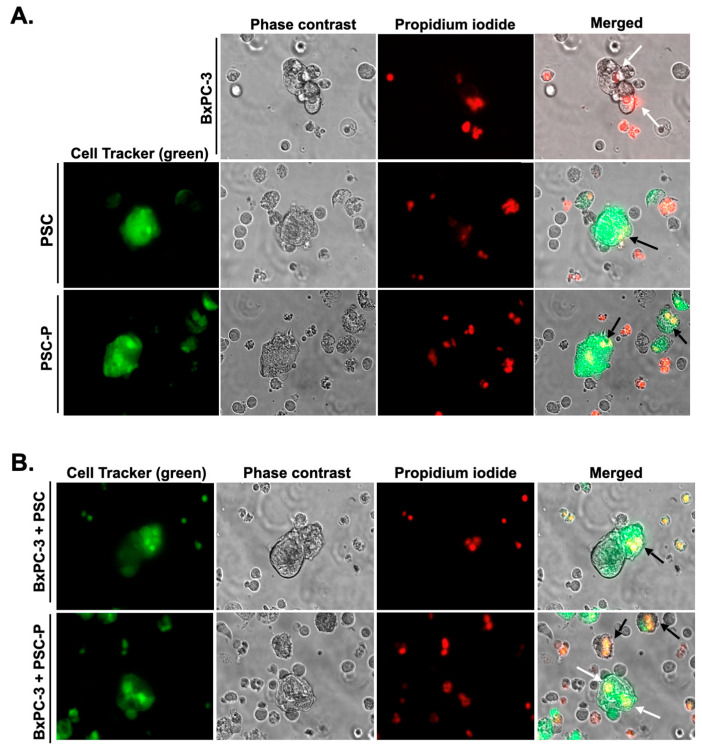
Increased PI posivity were observed in the spheroids formed by PSC-P cells. In order to observe cell death in PSC and PSC-P cells, the cells were stained with CellTracker green and then cultured alone (**A**) or with non-stained BxPC-3 cells (**B**) for 4 days to form spheroids. Following 24 h exposure to NK-92 cells, the cells were stained with PI. White arrows show cell death in BxPC-3 cells, while black arrows represent cell death in PSC and PSC-P cells.

## Data Availability

The data presented in this study are available on the request from the corresponding author.

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
