# Peer review of "Targeting Periostin Expression Makes Pancreatic Cancer Spheroids More Vulnerable to Natural Killer Cells"

_biomedicines, 2023, doi:10.3390/biomedicines11020270_

Round 1
Reviewer 1 Report
This manuscript titled as “Targeting pancreatic stellate cells-derived periostin makes the pancreatic cancer spheroids more vulnerable to natural killer cells”, is submitted by Didem Karakas, et al. In this study, the authors used CRISPR/Cas 9 technology to knockdown the periostin expression in PSC cells and demonstrated that it decreased aSMA and collagen expression level. The authors then demonstrated that targeting periostin led to the infiltration and activation of NK cells in co-cultured system. In general, it is a neat story. Although periostin in PSC has been previously described but its association with NK cell infiltration stays novel. The application of primary CAFs rather than normal PSCs could significantly improve the rigor of the study.
Minor
1. Figure 1D, to my knowledge, KmPltter, Kaplan-Meier plotter [Gastric] (kmplot.com), does not contain PDAC database. Please attach details of patient information that generated figure 1d or attach the correct hyperlink.
2. Figure 4a does not differentiate cell death of BxPC3 or PSC without co-staining data from EpCam.
3. Line 394-7, “PI positivity increased in PSC-P mono-culture spheroids compared to the spheroids formed by PSC cells (Figure 4B). A similar pattern was observed in co-culture spheroids formed by BxPC-3+PSC and BxPC-3+PSC-P cells (Figure 4C).” these data is not convincing and also needs more explicit description.
Author Response
1. Figure 1D, to my knowledge, KmPltter, Kaplan-Meier plotter [Gastric] (kmplot.com), does not contain PDAC database. Please attach details of patient information that generated figure 1d or attach the correct hyperlink.
Response: We chose the patients with pancreatic ductal adenocarcinoma to analze POSTN mRNA expression based on RNA-seq data. The link of the analysis given below:
https://kmplot.com/analysis/index.php?p=view&pa_id=16946452&show=bZFNboQwDIXvwrqqNItuehnLBGewCHFkO8y0o969gUULiO33np__Xp3hQoAGzjF2n13EZNS9dTbKA2YaGHOjrvUfcp7xeYI9kT1Q5-sEkAhWdeEF07kQw9QKB1CKpJQDnSNKEofEmcBHDlMms2a5HWSnp0MvabhWjL_X2Nv7x0EL1VxmmJFzW9_TdevShq4ERYydZbuGFOX76McJ6U552Nt68RafKK7GolLCSGHa9fhjgCk98Mt2WjWCBZWxTwRRFIKMor63BFEFK9TOvj3pwFdsks8bheqy_Xn7wc8v
2. Figure 4a does not differentiate cell death of BxPC3 or PSC without co-staining data from EpCam.
Response: The flow cytometry analysis graphs showing Annexin V and PI positivity in total cell population (in both BxPC-3 and PSC/PSC-P) were added as supplementary figure 3.
3. Line 394-7, “PI positivity increased in PSC-P mono-culture spheroids compared to the spheroids formed by PSC cells (Figure 4B). A similar pattern was observed in co-culture spheroids formed by BxPC-3+PSC and BxPC-3+PSC-P cells (Figure 4C).” these data is not convincing and also needs more explicit description.
Response: In the present study, our goal is to evaluate the changes in cancer cells' death depending on collagen changes. Therefore, we did not focus on cell death in PSCs in detail. Besides, as we mentioned in the manuscript, we could not perform flow cytometry analysis for Annexin-V and PI in PSCs because the marker used to gate the cells requires a permeabilization step which causes false positivity for annexin and PI staining. Briefly, here we aimed to produce general information about periostin manipulation in PSCs and its relationship with NK cell cytotoxicity and, thus cell death in cancer cells. As a future project, we have been planning to perform some detailed analysis to dissect the points that remain unclear.
Reviewer 2 Report
Karakas and co-authors submitted an article on Targeting pancreatic stellate cells-derived periostin makes the pancreatic cancer spheroids more vulnerable to natural killer cells. Pancreatic stellate cells (PSCs) are the most abundant cells in the pancreatic cancer (PaCa) microenvironment and the principal source of collagen. Periostin is specifically produced by PSCs and promotes the aggressiveness of PaCa cells by facilitating extracellular collagen assembly. There is clear scientific merit in study that periostin might be a therapeutic target for PaCa. However, in its present stage, the manuscript presents deficiencies that should be revised to reach the possibility of publication.
1. Overall, all figures are missing scale bars.
2. In Figure 2C, it is recommended to quantify invasion ability after performing invasion assay using transwell.
3. In Figure 3C-E and 4A, Quantitative graphs with statistics are required.
4. Would it be possible to add a section with a good discussion about clinical treatments or in vivo. in tests or already being applied, that target Periostin in PSCs?
Author Response
1. Overall, all figures are missing scale bars.
Response: Added in the related figures.
2. In Figure 2C, it is recommended to quantify invasion ability after performing invasion assay using transwell.
Response: Transwell invasion assay can be performed as you suggested. However, considering the time period given for revision (10 days) and the time we need to spend on the experiment, it does not seem possible to complete this experiment in 10 days. The proliferation ability of PSC cells is dramatically decreased after periostin knockdown, thus it takes a long period of time to culture them. Then, at least 3 more days are needed for the spheroid formation and 1 day for invasion assay, and 1 day for analysis and quantification. Therefore, unfortunately, we could not complete this experiment. If you give us more time, we can complete this experiment.
3. In Figure 3C-E and 4A, Quantitative graphs with statistics are required.
Response: Quantitative graphs were added in the related figures.
4. Would it be possible to add a section with a good discussion about clinical treatments or in vivo. in tests or already being applied, that target Periostin in PSCs?
Response: Based on the detailed literature search, there is no clinical trial targeting periostin. On the other hand, a few in vivo study findings were added and highlighted in discussion section. Besides, there are some strategies to target stroma of pancreatic cancer evaluating in clinical trials were added and highlighted in discussion.
Round 2
Reviewer 2 Report
The author responded appropriately to my comment.
The paper will be of new insight into the pancreatic cancer microenvironment field and these researchers.
Author Response
Thank you for your valuable comments.
Kind regards,